

# Effects of microhabitat on rodent-mediated seed removal of endangered *Kmeria septentrionalis* in the karst habitat

Guohai Wang[1,*], Yang Pan[2,*], Guole Qin[3], Weining Tan[4] and Changhu Lu[1]

[1] College of Biology and the Environment, Nanjing Forestry University, Nanjing, Jiangsu, China
[2] Nanjing Institute of Environmental Sciences, Ministry of Ecology and Environment, Nanjing, Jiangsu, China
[3] College of Chemistry and Bioengineering, Hechi University, Yizhou, Guangxi, China
[4] Management Bureau of Mulun National Nature Reserve, Nature, Huanjiang, Guangxi, China
[*] These authors contributed equally to this work.

## ABSTRACT

Seed removal behaviors of rodents are largely influenced by microhabitat. Although the karst ecosystem is composed of a broad variety of microhabitats, we have no information on how they affect such behaviors. We investigated rodents' seed removal behaviors in four karst microhabitats (stone cavern, stone groove, stone surface, and soil surface) using three types of *Kmeria septentrionalis* seeds: fresh, black (intact seeds with black aril that dehydrates and darkens), and exposed (clean seeds without the aril). We show that *Rattus norvegicus*, *Leopoldamys edwardsi* and *Rattus flavipectus* were the predominant seed predators. Even though all seed types experienced a high removal rate in all four microhabitats, but rodents preferentially removed seeds from the three stone microhabitats (stone caves: $69.71 \pm 2.74$; stone surface: $60.53 \pm 2.90\%$; stone groove: $56.94 \pm 2.91\%$) compared to the soil surface ($53.90 \pm 2.92\%$). Seeds that had been altered by being exposed to the environment were more attractive to rodents than fresh seeds ($76.25 \pm 2.20\%$ versus $36.18 \pm 2.29\%$). The seed removal behavior of rodents was significantly affected by the microhabitat and seed type. Finally, seeds that had fallen on the soil surface microhabitat incurred a lower predation risk than seeds fallen on other microhabitats, which increased their probability to germinate. Our results indicate that the lower predation rate of seeds from the endangered *K. septentrionalis* dropped on the soil surface increases trees' likelihood of survival.

# INTRODUCTION

Seed removal by rodents is one of the most important dispersal mechanisms for many plant species (*Lichti, Steele & Swihart, 2017*; *Wang & Corlett, 2017*). They tend to transport seeds away from the mother tree, where they would otherwise experience strong parental competition (*Nathan & Muller-Landau, 2000*; *Jansen et al., 2014*). This behavior also increases the chances of seeds reaching a suitable germination habitat and facilitates their distribution (*Steele et al., 2015*; *Wang & Corlett, 2017*). The pattern and intensity of seed

Corresponding author
Changhu Lu, luchanghu@njfu.com.cn

removal depends on a multitude of biotic and abiotic factors, such as seed traits and availability (*Dylewski et al., 2020*; *Xiao, Zhang & Krebs, 2015*), rodent abundance (*Li & Zhang, 2007*) and habitat characteristics (*Wang et al., 2019b*; *Perea et al., 2012*). A seed's microhabitat is one of the most important factors affecting its removal (*García-Castaño, Kollmann & Jordano, 2006*; *Steele et al., 2015*).

Microhabitats provide a variety of biological and abiotic conditions for the survival of rodents (*Vander-Wall, 2000*; *Fleury & Galetti, 2006*) and determine the quality of the foraging habitat through predation risk and foraging costs. A microhabitat's characteristics influence the abundance, activity intensity, and spatial distribution of rodents (*Pérez-Ramos & Marañón, 2008*), ultimately affecting the probability of seed encounters and foraging behaviors of seed predators (e.g., removal or in situ consumption; *Perea et al., 2012*; *Reed, Kaufman & Kaufman, 2005*). For example, *Peromyscus polionotus* removed more seeds from dense vegetation cover than from open patches where there was a higher possibility of being confronted by predators (*Orrock, Danielson & Brinkerhoff, 2004*). The foraging behavior of rodents with respect to seeds in different microhabitats affects seedling regeneration, spatial distribution, and the diversity of trees (*Hirsch et al., 2012*; *Zhang et al., 2016*). *Steele et al. (2014)* found that eastern gray squirrels (*Sciurus carolinensis*) dispersed larger, more profitable acorns (*Quercus spp.*) into open habitats, with a lower rate of cache pilferage, and better conditions for seedling establishment. The seed removal behavior of rodents is also affected by changes in seed presentation (cleaned seeds versus seeds within the pulp) (*Perea, San & Gil, 2011*; *Pan et al., 2016*). Some studies have shown that rodents prefer to remove exposed seeds that have been regurgitated or defecated by birds versus intact seeds (*Perea, San & Gil, 2011*).

The karst habitat of southwest China represents one of the world's most spectacular examples of tropical-subtropical karst landscapes. It is characterized by high edaphic and topographic heterogeneity formed by several types of microhabitats (e.g., stone groove, stone crevice, and stone cavern) (*Clements et al., 2006*; *Zhang et al., 2013*). The plants in the karst habitat bear a large amount of seeds every year (*Pan, Nai & Li, 2008*; *Tang et al., 2011*), but it is extremely difficult for them to spread over long distances due to geographic barriers and most of them end up randomly falling into various microhabitats around the mother tree. Therefore, the seeds of many plants in the karst habitat must rely on ground-dwelling rodent removal to a suitable microhabitat for germination. However, there are few studies on the behavior of rodent seed predators in the karst habitat.

*Kmeria septentrionalis* is a dioecious tree species of the Magnoliaceae with a red aril that is endemic to China and is listed as an endangered species by the IUCN. It is a first-class national protected plant in China (*Lin et al., 2011*). The seeds rely on birds for their dispersal and become completely exposed after passing through bird's digestive system, which results in them having no pulp (*Wang et al., 2019a*). Seeds that are not removed by birds fall to the ground and remain intact (fresh seeds) and the red aril of fresh seeds dehydrates and turns black (black seeds) after approximately $138.20 \pm 3.86$ h ($n = 30$). The topographic heterogeneity of karst habitat forms several types of microhabitats (e.g., stone groove, stone crevice, and stone cavern) (*Clements et al., 2006*; *Zhang et al., 2013*) and the

three types of *K. septentrionalis* seeds (fresh seeds, black seeds, and exposed seeds) are often randomly distributed in different microhabitats under the mother trees.

We conducted a rodent-mediated seed removal experiment in four karst microhabitats (stone cavern, stone groove, stone surface, and soil surface) with three types of *K. septentrionalis* seeds (fresh seeds, black seeds, and exposed seeds) to determine the effect of the microhabitat on the seed removal behavior of rodents. We had two objectives: (1) how karst microhabitats affect seed removal behaviors from rodents predating on seeds of *K. septentrionalis*; (2) to determine whether rodents preferred a specific seed type. We predicted that the seed removal rate in the stone microhabitats was higher than in the soil surface, and fresh seeds had lower removal rate than other types.

## MATERIALS & METHODS

### Ethics statement
Field studies were conducted under the permission from the Administrative Bureau of Mulun National Nature Reserve. The Institutional Animal Care and Use Committee at College of Biology and the Environment, Nanjing Forestry University approved this research.

### Study area
We performed field experiments in the Mulun National Nature Reserve (107°54′01″– 108°05′51″E; 25°07′01″– 25°12′22″N) in Guangxi Zhuang Autonomous Region, southwest of China (Fig. 1). The nature reserve has typical karst landforms consisting of peak-cluster depressions and valleys, and altitudes ranging from 300–1000 m above sea level. The climate in this region is dominated by the mid-subtropical zonal monsoon and the averages annual air temperature is approximately 19.3 °C, ranging from −5 °C in January to 26.7 °C in July. The annual rainfall averages 920 mm with the highest amounts of rainfall occurring from June to September (*Liu et al., 2012*). The annual frost-free period lasts approximately 235–290 days, and the relative humidity is typically higher than 79% (*Pan, Nai & Li, 2008*). The local vegetation dominated by middle subtropical evergreen and deciduous broad-leaved mixed forest, including species liked *K. septentrionalis, Lindera communis, Machilus pingii*, and *Loropetalum chinense* (*Wang et al., 2019a*).

### Study species
*Kmeria septentrionalis* is classified as an endangered species by the IUCN (*Lin et al., 2011*) and is only found in the karst habitat of the Guangxi Zhuang Autonomous Region (Luocheng, Huanjiang), Guizhou Province (Libo) and Yunnan Province (Malipo, Maguan). Most plants are found in the Mulun National Nature Reserve (*Pan, Nai & Li, 2008*). Female plants bear 100–300 fruits annually and each fruit contains 4–14 seeds (mean ± standard error, length, 1.14 ± 0.15 cm; width, 0.49 ± 0.06 cm and weight, 0.23 ± 0.03 g; $n = 30$) (*Wang et al., 2019a*). The fleshy arils turn red in autumn and attract birds to forage and disperse the seeds. *Hemixos castanonotus, Yuhina castaniceps*, and *Pericrocotus flammeus* are the main seed dispersers, and these birds consume large

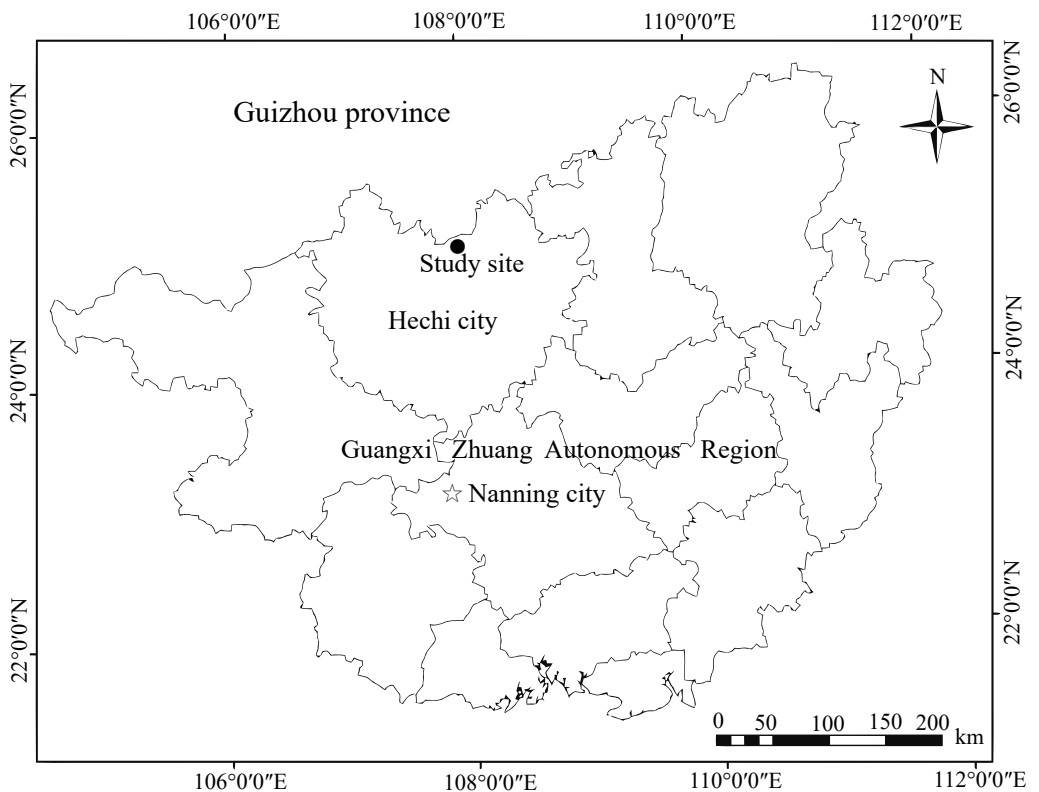

**Figure 1** **Location map of Mulun National Nature Reserve, China.**

numbers of seeds, which are excreted to various microhabitats after passing through the birds' digestive tracts (*Wang et al., 2019a*).

## Rodent species survey

We baited live traps (25 cm × 11 cm × 11 cm; Power of Arrest, China) with peanuts and fresh *K. septentrionalis* seeds to determine the species of rodents under the canopies of the mother forest during the mature period of *K. septentrionalis* seeds. The trapping stations were set approximately 5 m apart to minimize the effects of trapping in microhabitats where the seeds were placed. Three 30 m transects were selected according to the distribution of the mother trees, and 5 trap stations were set at 5 m intervals each transect for ten consecutive days. 150 traps were set in total. Traps were left open and were checked daily at 7:00 am and 7:00 pm. We recorded the species and body weights of the captured rodents, which were released immediately after feeding on different types of *K. septentrionalis* seeds. Six infrared cameras (Loreda L710, Leyueda Electronics Co., Ltd. Shenzhen, China) were set at 10 m intervals to monitoring the rodent species. The cameras were set to take three photos and one video after each trigger. The time interval between each trigger was 5 s. All cameras were operated for 24 h a day over ten consecutive days.

**Table 1  Description of karst microhabitats in the study area.**

| Microhabitat | Description |
| --- | --- |
| Stone cavern | the bedrock vertical sunken to form a semi closed cave |
| Stone groove | the bedrock protrudes horizontally to form a semi-closed strip-like fissure |
| Stone surface | a small tableland with more than 30% of the bedrock exposed |
| Soil surface | a continuous soil surface with a length and width greater than 2 m |

## Selection of microhabitats for seed placement

Stone cavern, stone groove, stone surface, and soil surface were the most representative karst microhabitats found in the study area. The microhabitats were distributed under the tree canopies of *K. septentrionalis* trees. We selected these four karst microhabitats as seed placement sites to study the seed removal rates of *K. septentrionalis* seeds. The specific definitions of these microhabitats are description by *Lu et al. (2010)* (Table 1, Fig. 2).

## Seed removal experiment

Seed removal experiments were conducted in late September to mid-October 2018 during the natural maturity season of *K. septentrionalis* seeds. Three types of *K. septentrionalis* seeds were placed in the four microhabitats. Seeds with intact fleshy arils that untouched by animals were referred to as 'fresh seeds', and were collected from the ground or directly from different mother trees. Black seeds, referred to as 'intact seeds', had a black aril that had not been touched by animals. These were collected from the ground or created by placing intact fresh seeds on the ground for approximately $138.20 \pm 3.86$ h ($n = 30$). Exposed seeds, referred to as 'clean seeds', after passed through the digestive tract of birds and had no aril or pulp. It was difficult to collect a statistically significant number of exposed seeds on the ground because of the high heterogeneity of the karst habitat so we created clean seeds by carefully extracting them from the fresh seeds. We wore plastic gloves when collecting the *K. septentrionalis* seeds and preparing the experimental apparatus to avoid contamination.

Three types of seeds were placed in each microhabitat in three plastic Petri dishes (diameter, 90 mm), which were placed at intervals of at least 10 cm. Thirty seeds of each type were placed in each Petri dish (30 fresh seeds, 30 black seeds, 30 exposed seeds). 90 seeds in total were placed in each microhabitat. We set up seven stations for each microhabitat every day, totaling twenty-eight stations, with an interval of 10 m between two stations to ensure the independence of experimental units. We dismantled and re-established all the stations randomly every day to avoid any spatial pseudoreplication and the experiments were carried out over 10 consecutive days. Our experimental set-up consisted of 280 microhabitat stations, 840 plastic Petri dishes, and 25200 seeds ($30 \times 3 \times 28 \times 10$). Seeds were placed on Petri dishes in the morning and left for 24 h. The state of the seeds and data were checked and recorded daily at (0700–0900 h). The remaining seeds were removed and replaced with new ones. During the field observations, it was noted that ants do not remove

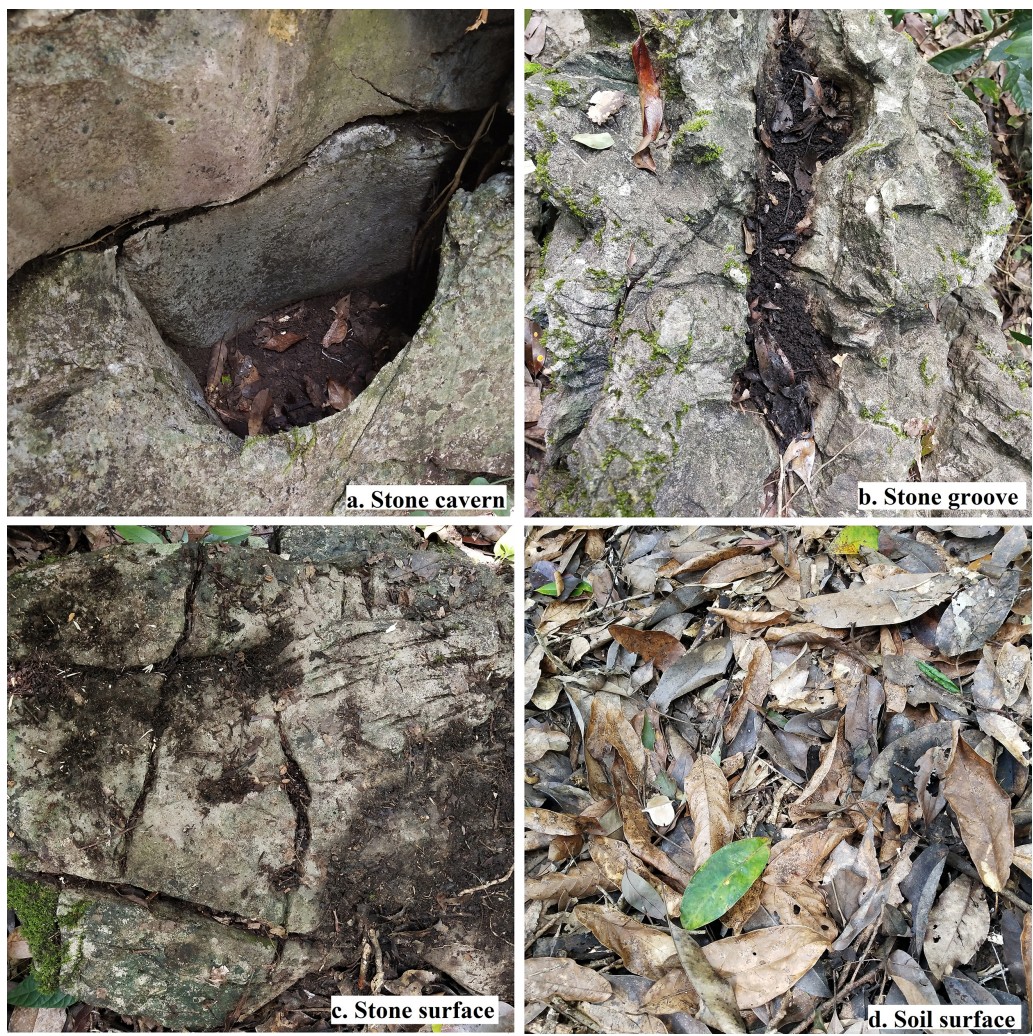

**Figure 2** **Photograph of four karst microhabitats in the study area.** (A) Stone cavern. (B) Stone groove. (C) Stone surface. (D) Soil surface.

the seeds of *K. septentrionalis*, but they fed on the elaiosome of seeds in situ. We were able to confirm that the seeds of *K. septentrionalis* were consumed or removed by rodents based on trapping, infrared camera monitoring, and subsequent feeding trials in cages. We did not record the final state of the seeds but we considered the seeds removed by rodents if they (i) were missing from the plastic Petri dishes; or (ii) were still on the dishes but were gnawed and empty (*García, Martínez & Obeso, 2007*; *Pan et al., 2016*). The removal rate was calculated as the proportion of removed seeds relative to the initial number of seeds, and the average removal rate was taken as the removal rate for each microhabitat and seed type.

## Statistical analysis

Mann–Whitney U tests were used to compare the differences in seed removal rates between the two microhabitats and seed types, respectively. Generalized linear mixed
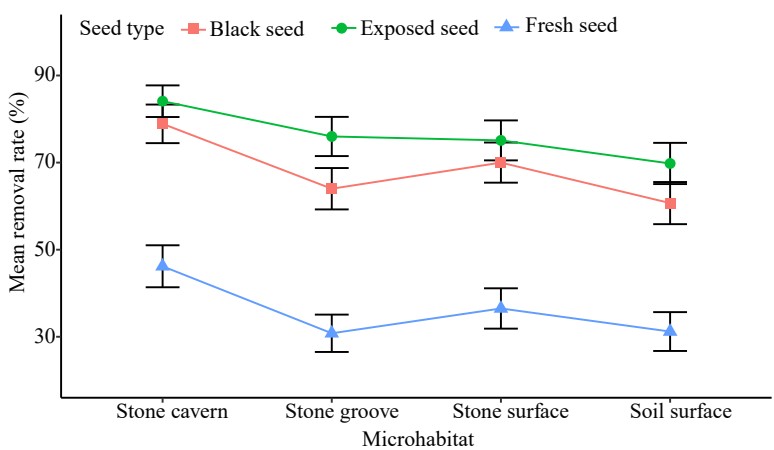

**Figure 3 The seed removal rates in different microhabitats and seed types.**

models (GLMM; lme4 package, version 3.2.5, *R Core Team, 2016*) were used to estimate the effect of microhabitats and seed types on the seed removal rate with the day of the experiment and the microhabitat station ID set as random factors. All data analysis and figure creation were performed using the R program, and the level of statistical significance was set at $P < 0.05$.

## RESULTS

### Rodent species

9 rodents were trapped, and the trap success was estimated at 6.00%, representing a low level of rodent abundance. *Rattus norvegicus* made up 55.56% ($n = 5$), *Leopoldamys edwardsi* made up 33.33% ($n = 3$), and *R. flavipectus* made up 11.11% ($n = 1$) of the total number of rats trapped. A total of 548 photos and 174 videos were taken by the infrared camera, of which 362 pictures and 69 videos contained rodents. All of the rodents captured on camera were unique members of the three types of rodents. The seed traces were the same in the captured rodents as those in the field, and infrared cameras recorded videos showing these rodents were the main predators of *K. septentrionalis* seeds.

### Seed removal

There was a relatively high rate of seed removal in all of the microhabitats. The average seed removal rate in three of the stone microhabitats was perceptibly higher than that of the soil surface microhabitat $53.90 \pm 2.92\%$ (stone cavern: $69.71 \pm 2.74\%$; stone surface: $60.53 \pm 2.90\%$; and stone groove: $56.94 \pm 2.91\%$). Significant differences were noted in the seed removal rate between the stone cavern and other microhabitats ($P < 0.001$), respectively (Fig. 3).

Exposed seeds had the highest average removal rate across all microhabitats (69.8%–84.1%), and fresh seeds had the lowest removal rate (30.8%–46.2%) (Fig. 3). The removal rates of fresh seeds were significantly different from those of black seeds and exposed seeds in all microhabitats ($P < 0.001$). The removal rates of black seeds and exposed seeds were

**Table 2  Results of generalized linear mixed models (GLMM) evaluating the effects of microhabitats and seed types on the rodent-mediated seed removal rate.**

| Variable | Estimate | Standard error | $t$-value | $P$-value |
|---|---|---|---|---|
| Intercept | 0.456 | 0.059 | 7.799 | 0.000 |
| Microhabitat | −0.226 | 0.089 | −2.523 | 0.012 |
| Seed type | 0.871 | 0.113 | 7.686 | 0.000 |
| Microhabitat*Seed type | −0.019 | 0.275 | −0.068 | 0.946 |

only significantly different in the stone groove microhabitat ($P = 0.013$). Furthermore, the seed removal rate was significantly affected by both microhabitat and seed type (Table 2). The seeds placed in stones caves and the exposed seeds seemed to attract rodents because they had the highest removal rates.

## DISCUSSION

We found that all types of seeds in the four microhabitats had a high removal rate by rodents (Fig. 3), which is similar to other studies that also reported such high rate of seed removal by rodents (*Vander-Wall, 2003*; *Pan et al., 2016*; *Li & Zhang, 2007*). Only nine rodents were captured, representing a low level of rodent abundance, which may be due to the close proximity of the study area to a village. A large number of *Ipomoea batatas*, *Oryza sativa* and *Glycine max* were planted in nearby farmlands during the mature period of the *K. septentrionalis* seed. The small size and low nutrient content of *K. septentrionalis* seed encourages the migration of rodents to nearby farmland to obtain enough food to meet their daily energy needs, reducing the density of the rodent population in the forest. Previous studies have shown that *L. edwardsi*, *R. norvegicus*, and *R. flavipectus* tend to cache seeds for later use in periods of food scarcity (*Chang, Xiao & Zhang, 2010*; *Shepherd & Ditgen, 2013*; *Cao, Yan & Wang, 2018*). Rodents prefer to disperse and cache large seeds with high nutritional value and will consume small seeds immediately to compensate for energy expenditure during foraging (*Chang, Xiao & Zhang, 2009*; *Cao, Yan & Wang, 2018*). The medium size of *K. septentrionalis* seeds have a lower nutrient content (*Zheng, 2016*), and the rodents needed to consume more seeds to ensure sufficient nutrition. Many seed fragments were left around the Petri dishes, leading us to speculate that most of these seeds were probably eaten in situ by rodents and not removed and cached. Similar results have been found in other studies of fleshy fruits (*García, Obeso & Martínez, 2005*; *Pan et al., 2016*). It has been suggested that the seed fragments discarded by rodents may germinate and develop seedlings (*Loayza et al., 2014*), however, when we followed up after a year there were no successfully germinated of the seed fragments discarded by rodents, indicating that rodents may be predators to the seeds of *K. septentrionalis*.

The seed removal rates in stone microhabitats were significantly higher than those on the soil surface (Fig. 3). This pattern could be attributed to the foraging behavior of small rodents, which are more active in sheltered microhabitats (*Pérez-Ramos & Marañón, 2008*). Stone microhabitats may provide a refuge for rodents, reducing their exposure time, and the risk of being caught by predators. This was especially true in the stone cavern

where the risk of being caught by large carnivorous predators was reduced because of the complex and narrow space, and darkened conditions are more conducive to rodent hiding (*Vander-Wall, 2000*). However, the high seed removal rates in stone microhabitats may not facilitate regeneration for this endangered plant species. This is not only because habitat conditions, such as shallow soils, and low nutrient and water contents, do not create a conducive environment for *K. septentrionalis* seed germination and seedling growth, but also because the rodents that were fed *K. septentrionalis* seeds in situ do not leave any behind to germinate. Conversely, seeds fallen on the soil surface incurred a lower predation risk, and we found that almost all the seedlings were grown in the soil surface when investigated the establishment of *K. septentrionalis* populations in fields (File S1). Therefore, we deduced that the seeds that fallen on the soil surface microhabitat may survive and that the soil surface might be more beneficial to the establishment of plants than other karst microhabitats.

The probability of seeds being removed varied by the type of seed, and fresh seeds had the lowest removal rate in all microhabitats (Fig. 3), which supports the findings of other studies (*Perea, San & Gil, 2011*; *Pan et al., 2016*). The intra-specific differences in seed removal rates may be related to secondary metabolites in these seeds. Fresh seeds of *K. septentrionalis* contained large amounts of volatile monoterpenoids and possibly had poor palatability (*Huang et al., 2010*), which may reduce the interest in seed removal by rodents. However, the seeds become dehydrated and turn black after falling to the ground, and it is possible that the concentration of some of the unpalatable secondary substances may decrease during this process, increasing the rate of seed removal. Additional studies should focus on the secondary substances involved their influence on the removal of *K. septentrionalis* seeds by rodents. We found that rodents always use fruit-handling methods and bit into the aril to feed on the seed kernel, rather than removing or consuming the whole seed. Exposed seeds, which are easier for rodents to manipulate than intact seeds, were favored, reflecting a foraging behavior that involves acquiring the most energy with the least input of time and energy and the lowest predation risk (*Fedriani & Manzaneda, 2005*). Exposed seeds had the highest removal rate across all microhabitats (Fig. 3). Other studies also found that the removal rate of exposed seeds was greater than that of intact seeds (*Perea, San & Gil, 2011*).

## CONCLUSIONS

Our study suggested that the seed removal rate by rodents was significantly affected by the various karst microhabitats and the seed types of *K. septentrionalis*. The seed removal rates in stone microhabitats were significantly higher than on the soil surface microhabitat. Rodents preferred to remove seeds in stone caves and exhibited a preference for exposed seeds. Therefore, we determined that the seeds dropped on the soil surface face a lower predation rate and will increase the survival of the endangered *K. septentrionalis*.

## ACKNOWLEDGEMENTS

We thank the staff of the Mulun National Nature Reserve for their contributions in the field.

### Funding

This work was supported by the National Natural Science Foundation of China (No.30970470) and Jiangsu Planned Projects for Postdoctoral Research Funds (No. 2018K064B). The funders had no role in study design, data collection and analysis, decision to publish, or preparation of the manuscript.

### Grant Disclosures

The following grant information was disclosed by the authors:
National Natural Science Foundation of China: 30970470.
Jiangsu Planned Projects for Postdoctoral Research Funds: 2018K064B.

### Competing Interests

The authors declare there are no competing interests.

### Author Contributions

- Guohai Wang and Yang Pan conceived and designed the experiments, performed the experiments, analyzed the data, prepared figures and/or tables, authored or reviewed drafts of the paper, and approved the final draft.
- Guole Qin analyzed the data, prepared figures and/or tables, and approved the final draft.
- Weining Tan performed the experiments, prepared figures and/or tables, and approved the final draft.
- Changhu Lu conceived and designed the experiments, analyzed the data, authored or reviewed drafts of the paper, and approved the final draft.

### Animal Ethics

The following information was supplied relating to ethical approvals (i.e., approving body and any reference numbers):

The Institutional Animal Care and Use Committee at the College of Biology and the Environment, Nanjing Forestry University approved this research.

### Field Study Permissions

The following information was supplied relating to field study approvals (i.e., approving body and any reference numbers):

Field experiments were approved by the Administrative Bureau of Mulun National Nature Reserve.

### Data Availability

The raw measurements are available in the Supplementary Files.

## Supplemental Information

Supplemental information for this article can be found online at http://dx.doi.org/10.7717/peerj.10378#supplemental-information.

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
