# Peer review of "Effects of microhabitat on rodent-mediated seed removal of endangered Kmeria septentrionalis in the karst habitat"

_PeerJ, doi:10.7717/peerj.10378_

## Round 0.1 · original submission · Major Revisions

I have read the reviews and the manuscript myself and came to the same conclusion as the reviewers that the statistical analyses need to be thoroughly revised.

Both reviewers suggest mixed model, please justify your approach (although I have seen that Petri dish location change every day to avoid pseudoreplication).

Also, I don't understand why you have conducted correlation analyses. Why would you test for a correlation between your categorical variables?

Fig 3 needs to be redrawn, and figure 4 and 5 should relate to the glm, not correlations. Label your axes appropriately and provide more details in the legend. What were the red lines in your graph? Strong correlation such as 0.78 doesn't seem to be supported by the associated graph.

Please make sure that all statistics are carefully reviewed and make sure you address all comments provided by the reviewers.

Regards,

Patrick

·

Basic reporting

I enjoyed reading your manuscript and overall, I am of the opinion that it is well written and your ideas are easy to follow.

1. Background and goals (abstract and introduction)

The abstract should be re-written, with more background provided, namely regarding karst ecosystems, their specificity or the need to study them. In addition, clear goals are lacking; it seems that your goal were to improve your knowledge about seed removal in these ecosystems, but you should be more specific: did you want to determine the dominant rodent species removing the seeds? The causes and consequences of seed removal? Microhabitat differences in rates of seed removal? This should be clear by reading the abstract.

Introduction
I think that some concepts need to be introduced with more details. For instance L.51-58 you refer as changes in seed morphology. When you do this, the readers first think of seed size or shape. But here, you refer to a much broader phenomenon; seeds do not only change in size or shape when going through the digestive system of birds, but also in composition, nutritional qualities, odor... and this is not morphology. I suggest that you re-write these sections of your introduction and make it clearer for the reader; and also that you focused only on shape/size and not the other processes.

In addition, you should also better introduced kast ecosystems here as well.

Finally, I think that you should more clearely expose your goals and perhaps make some predictions at the end of your introduction (see detailed comments on the experimental design section).

2. Raw data
I thank you for sharing your raw data. However, there are not sufficient information to perform analyses with this or to be able to assess the validity of your analyses. You should provide the dataset on which you conducted your GLM, with your response variable in column and a column for each of your factors (type of microhabitat, type of seeds, site, day…).

3. Figures
I think that the figures and especially the type of figure for Fig.3 should be revised, as it is difficult to assess the differences with this representation of the data. I make some specific suggestions in the pdf. You should also provide more information in the caption (see detailed comments in the pdf)

4. Minor comments

Although well writtent, there aare still some sentences or ideas that would benefit to be rephrased; see examples L108-109, 148-150. I have made specific comments for each of them within the pdf.

I found the title quite long, perhaps “in the karst habitat, Guangxi, southwestern China » is not mandatory?

The literature was well referenced and relevant.
One suggestion for your introduction: Dylewski, Ł., Ortega, Y. K., Bogdziewicz, M., & Pearson, D. E. (2020). Seed size predicts global effects of small mammal seed predation on plant recruitment. Ecology Letters, 23(6), 1024-1033.

Experimental design

1. Statistical analyses

My main concern is about statistical analyses. I don't think that the Mann-Whitney or K-S tests are necessary since you conducted a GLM, which seems more appropriate for your analyses. Why did you perform both?

You should provide more details on your GLM (did you use a binomial or a poisson with an offset?). What were the variables that you included? And your response variable (number or proportion of seeds removed)? Consider providing the codes in supp info.

However, I also think that you should consider conducting a GLMM, and include the day of the experiment (since you have 10 days with repeated measurements) and the microhabitat station ID as random effects. Indeed, your data are non-independent and as such, the glm does not seem to be the most appropriate to analyze your data.

2. Although your overall goal is clearly stated and in line with the background provided, I think you should provide more details, in the abstract, but also in your introduction. For instance, L72-74, you could include some predictions. And you should be more specific: was your goal (1) to characterize seed removal (proportions, size, distance… of seeds removed, species targeting the seeds) or to measure the effects of seed type and microhabitat type on the proportion of seeds of an endangered species that were removed, with specific predictions that can be made? Both options are valid, but I think that it should be stated. Also because you conducted an experimental design, which most often allow to investigate clear hypotheses and test predictions.

3. Exposed seeds that you created

L129 – you "created exposed seeds”. But I assume that their pH, odor, texture and nutritional composition was not the same than actual "exposed seeds" that had passed through the digestive tracts of birds? Did you measure differences (i.e. conducted some comparative analyses between real exposed seeds and the ones you created)? Or at least conducted a small food preference experiment, to see whether your seeds were as palatable as real exposed seeds? If not, then I think your conclusion would need to be tempered as you cannot be sure that these seeds indeed mimic real exposed seeds.

4. Protocol of capture and ethics

L107-110: Did you leave the traps open from 7am to 7pm? And checked them only twice? Or do you leave them open all along (during the entire period of 10 days) and checked them twice a day? In any case, this would mean that rodents could spend 12 hours within a trap? Did you have any mortality?

In addition, it seems to me that only 9 rodents captured in 10 days and 10 nights of capture is really low. Do you have an explanation for that?

5. Detailed comments on the experimental design
I think that in different sections (protocol of capture, experiments and statistics), more details are needed. I have made detailed comments in the pdf.

Validity of the findings

Although I do think that this experiment will yield valid findings, that will improve the knowledge on seed dispersion of an endangered tree species, it is difficult to make a comment on the validity of the finding at that stage.

Indeed, since, to my opinion:
1. statistical analyses have to be re-performed with a mixed model
2. the Fig. 3 does not allow to visualy assess the validity of the results presented (for microhabitat differences for instance)
3. the rax data provided do not allow to perform an analysis and would need to be completed and reorganized

I cannot assess the validity of the findings nor the conclusion in details.

Additional comments

Dear authors,
I have enjoyed reading your manuscript and can assess the strong involvement needed to conduct such experiment on site. Although I have made many comments, I have tried my best to remain constructive and hope that this feedback will benefit your manuscript.
Best

Reviewer 2 ·

Basic reporting

The article represents an interesting work, however, the article needs English to improve.

Experimental design

no comments

Validity of the findings

The statistical analysis needs to improve. I suggest used the generalized linear mixed models with beta distribution when you calculate the proportion of removed seeds. As a random factors the authors should use the stations ID.
I do not understand why authors used to many different analyses, which analyzed the same research questions.

Additional comments

Authors should be more claryfe in the method section and imporve the intriduction and discusion section.
The authors should change the analyses using the beta distribution.
I have a problem to interpretated the correlations matrix why authors use correlation beteween categorical vairlabes with the contionous viaralbes (removal rate) ?

---

## Round 0.2 · Minor Revisions

I have decided to resend your manuscript to one of the reviewer and received positive comments. I would like you to address the minor comments raised by the reviewer and I believe that your paper will then be accepted as I did not see any major problems. Thank you for the thorough revision you have provided and I am looking forward to receive your revised manuscript soon.

·

Basic reporting

1. BASIC REPORTING
Clear, unambiguous, professional English language used throughout.

The English has been much improved. I have provided some minor comments and tried to copy-edit the abstract that has been substantially modified, although I am myself not a native english speaker.

I am convinced by the authors’ answers to the reviewers’ comments, and despite my suggestions on the abstract and some theoretical issues with parts of the discussions, all my comments are minors.

Intro & background to show context.

Abstract: thanks for clarifying your goals in the abstract. However, I believe that it need English copy-editing. I am not a native English-speaker myself but I have tried to suggest some modifications that you can find below.

L21-22 > “Seeds that fell on the soil surface incured a lower predation risk, which increased the probability for the germinated seeds.” Lower than what? Other microhabitats?

Seed removal behaviors of rodents are largely influenced by microhabitat. Although the karst ecosystem is composed of a broad variety of microhabitats, we have no information on how they affect such behaviors. We investigated rodents’ seed removal behaviors in four karst microhabitats (stone cavern, stone groove, stone surface, and soil surface) using three types of Kmeria septentrionalis seeds: fresh, black (intact seeds with black aril that dehydrates and darkens), and exposed (clean seeds without the aril). We show that Rattus norvegicus, Leopoldamys edwardsi and Rattus flavipectus were the predominant seed predators. In addition, both the microhabitat and seed type influenced rodents’ seed removal behavior.
Indeed, even though all seed types experienced a high removal rate in all four microhabitats, but rodents preferentially removed seeds from the three stone microhabitats (stone caves: 69.71 ± 2.74%; stone surface: 60.53 ± 2.90%; stone groove: 56.94 ± 2.91%) compared to the soil surface (53.90 ± 2.92%). Seeds that had been altered by being exposed to the environment were more attractive to rodents than fresh seeds (76.25 ± 2.20% versus 36.18 ± 2.29%). The seed removal behavior of rodents was significantly affected by the microhabitat and seed type. Finally, seeds that had fallen in the soil surface microhabitat incurred a lower predation risk than seeds fallen in other microhabitats, which increased their probability to germinate. Our results indicate that the lower predation rate of seeds from the endangered K. septentrionalis dropped on the soil surface increases trees’ likelihood of survival.

Introduction

L32 – remove “However,” which would make us think that dispersion has negative effects. You’re just adding more positive effects to those described in your previous sentence.

L40 – the foraging cost – remove “the” and add a “s” at cost.

L41-43 – consider rewriting, for instance: ultimately affecting the probability of seed encounters and foraging behaviors of seed predators (e.g., removal or in situ consumption; Perea et al., 2012; Reed et al., 2005).

L54-55 – formed by several types of microhabitats

L60-61 – suggestion of revision: However, there are few studies on the behavior of rodent seed predators in the karst microhabitat.

L67 – in approximately OR after approximately

L67-68 – karst habitat or karst habitats? Sometimes there is an s, sometimes not “as in the title”.

L74-75 – suggestion of revision: (1) how karst microhabitats affect seed removal behaviors from rodents predating on seeds of K. septentrionalis;

Literature well referenced & relevant.
NA

Structure conforms to PeerJ standards, discipline norm, or improved for clarity.
Ok

Figures are relevant, high quality, well labelled & described.
Yes, I appreciate the work of the authors in improving the figures, especially Fig 3, although for Fig.1 I think that the previous version was of better quality.

Raw data supplied (see PeerJ policy).
I thank you for sharing your raw data.

Experimental design

I am convinced by the authors' modifications to this section and by the experimental design. I only provide minor comments/suggestions.

Materials and Methods – detailed comments

L80-81: please provide under which number.

L155-156 – avoid the use of double brackets > (GLMM ; lme4 package, version
156 3.2.5, R Core Team, 2016)

L171 – was perceptibly > add “was”, and what do you mean by perceptibly? On the graphs but not significant?

Captions
Table 1 – Description of table 1 (replace describe by description)

Validity of the findings

Discussion

L186 – add a “s” at seed.

L187 – replace “a high” by “such high”

L191 – replace “lower” by “low”

L193 – remove “mother”

L193 - Leopoldamys edwardsi, Rattus norvegicus > replace by L. edwardsi and R. norvegicus after first mention in the manuscript and write it consistently throughout the text, to also be consistent with R. flavipectus.

L195-196 – suggested modifications (replace larger/higher/smaller by large/high/small, trait by value and cost by expenditure): Rodents prefer to disperse and cache large seeds with high nutritional value and will consume small seeds immediately to compensate for energy expenditure during foraging.

L197 – I do not agree with this statement, is it demonstrated in this species or a speculation? Seeds may be small but highly concentrated in nutrients. It is true, however, that, to meet the same energy input, they will have to eat more small seeds than large seeds. But this does not mean that small seeds have poor nutritional value (also considering that energy input is not the only parameter to take into account when talking about nutritional value). Perhaps consider rephrasing, or if you have a reference confirming this affirmation in the specific case of K. septentrionalis, please mention it.

L202-204 – perhaps I am misunderstanding here, but I feel that this is contradictory to what you state in your abstract, no?

L206 – in sheltered habitats ("in" is missing)

L208 – in the stone cavern

L209-210 – please develop a bit more here, as I am not convinced by this simple statement. Some predators have an excellent sight in dark conditions.

L214-215 - Conversely, seeds falling on soil surfaces have lower removal rates, which may reduce the seed predation risk by rodents > well, the fact that seeds have lower removal rates is not a cause of the reduction in seed predation, but a consequence for a lower predation, no? I don’t understand your point here, could you be more explicit?

L228 – remove “the” before rodents

L238 – than on the soil surface

L238-239 - remove “the” before seeds

L240 – of the endangered K. septentrionalis.

---

## Round 0.3 · accepted · Accept

Hi, thank you for your revision, I am happy to accept your manuscript for publication.

Regards

Patrick